# A Unified Framework for Data Poisoning Attack to Graph-based Semi-supervised Learning

**Xuanqing Liu**
Department of Computer Science
UCLA
xqliu@cs.ucla.edu

**Si Si**
Google Research
sisidaisy@google.com

**Xiaojin Zhu**
Department of Computer Science
University of Wisconsin-Madison
jerryzhu@cs.wisc.edu

**Yang Li**
Google Research
liyang@google.com

**Cho-Jui Hsieh**
Department of Computer Science
UCLA
chohsieh@cs.ucla.edu

## Abstract

In this paper, we proposed a general framework for data poisoning attacks to graph-based semi-supervised learning (G-SSL). In this framework, we first unify different tasks, goals and constraints into a single formula for data poisoning attack in G-SSL, then we propose two specialized algorithms to efficiently solve two important cases — poisoning regression tasks under $\ell_2$-norm constraint and classification tasks under $\ell_0$-norm constraint. In the former case, we transform it into a non-convex trust region problem and show that our gradient-based algorithm with delicate initialization and update scheme finds the (globally) optimal perturbation. For the latter case, although it is an NP-hard integer programming problem, we propose a probabilistic solver that works much better than the classical greedy method. Lastly, we test our framework on real datasets and evaluate the robustness of G-SSL algorithms. For instance, on the MNIST binary classification problem (50000 training data with 50 labeled), flipping two labeled data is enough to make the model perform like random guess (around 50% error).

## 1 Introduction

Driven by the hardness of labeling work, graph-based semi-supervised learning (G-SSL) [1, 2, 3] has been widely used to boost the quality of models using easily accessible unlabeled data. The core idea behind it is that both labeled and unlabeled data coexist in the same manifold. For instance, in the transductive setting, we have label propagation [1] that transfers the label information from labeled nodes to neighboring nodes according to their proximity. While in the inductive case, a graph-based manifold regularizer can be added to many existing supervised learning models to enforce the smoothness of predictions on the data manifold [4, 5]. G-SSL has received a lot of attention; many of the applications are safety-critical such as drug discovery [6] and social media mining [7].

We aim to develop systematic and efficient data poisoning methods for poisoning G-SSL models. Our idea is partially motivated by the recent researches on the robustness of machine learning models to adversarial examples [8, 9]. These works mostly show that carefully designed, slightly perturbed inputs – also known as adversarial examples – can substantially degrade the performance of many machine learning models. We would like to tell apart this problem from our setting: adversarial attacks are performed during the testing phase and applied to test data [10, 11, 12, 13, 14, 15], whereas data poisoning attack is conducted during training phase [16, 17, 18, 19, 20], and perturbations are

added to training data only. In other words, data poisoning attack concerns about *how to imperceptibly change the training data to affect testing performance.* As we can imagine, this setting is more challenging than testing time adversarial attacks due to the hardness of propagating information through a sophisticated training algorithm.

Despite the efforts made on studying poisoning attack to supervised models [16, 17, 18, 19, 20], the robustness of semi-supervised algorithms has seldom been studied and many related questions remain unsolved. For instance, are semi-supervised learning algorithms sensitive to small perturbation of labels? And how do we formally measure the robustness of these algorithms?

In this paper, we initiate the first systematic study of data poisoning attacks against G-SSL. We mainly cover the widely used label propagation algorithm, but similar ideas can be applied to poisoning manifold regularization based SSL as well (see Appendix 4.2). To poison semi-supervised learning algorithms, we can either change the training labels or features. For label poisoning, we show it is a constrained quadratic minimization problem, and depending on whether it is a regression or classification task, we can take a continuous or discrete optimization method. For feature poisoning, we conduct gradient-based optimization with group Lasso regularization to enforce group sparsity (shown in Appendix 4.2). Using the proposed algorithms, we answer the questions mentioned above with several experiments. Our contributions can be summarized as follows:

- We propose a framework for data poisoning attack to G-SSL that 1) includes both classification and regression cases, 2) works under various kinds of constraints, and 3) assumes both complete and incomplete knowledge of algorithm user (also called "victim").
- For label poisoning to regression task, which is a nonconvex trust region problem, we design a specialized solver that can find a global minimum in asymptotically linear time.
- For label poisoning attack to classification task, which is an NP-hard integer programming problem, we propose a novel probabilistic solver that works in combination with gradient descent optimizer. Empirical results show that our method works much better than classical greedy methods.
- We design comprehensive experiments using the proposed poisoning algorithms on a variety of problems and datasets.

In what follows, we refer to the party running poisoning algorithm as the attacker, and the party doing the learning and inference work as the victim.

## 2  Related Work

Adversarial attacks have been extensively studied recently. Many recent works consider the test time attack, where the model is fixed, and the attacker slightly perturbs a testing example to change the model output completely [9]. We often formulate the attacking process as an optimization problem [10], which can be solved in the white-box setting. In this paper, we consider a different area called *data poisoning attack*, where we run the attack during training time — an attacker can carefully modify (or add/remove) data in the training set so that the model trained on the poisoned data either has significantly degraded performance [18, 16] or has some desired properties [21, 19]. As we mentioned, this is usually harder than test time attacks since the model is not predetermined. Poisoning attacks have been studied in several applications, including multi-task learning [20], image classification [21], matrix factorization for recommendation systems [19] and online learning [22]. However, they did not include semi-supervised learning, and the resulting algorithms are quite different from us.

To the best of our knowledge, [23, 24, 25] are the only related works on attacking semi-supervised learning models. They conduct **test time attacks** to Graph Convolutional Network (GCN). In summary, their contributions are different from us in several aspects: 1) the GCN algorithm is quite different from the classical SSL algorithms considered in this paper (e.g. label propagation and manifold regularization). Notably, we only use feature vectors and the graph will be constructed manually with kernel function. 2) Their works are restricted to testing time attacks by assuming the model is **learned and fixed**, and the goal of attacker is to find a perturbation to fool the established model. Although there are some experiments in [24] on poisoning attacks, the perturbation is still generated from **test time attack** and they did not design task-specific algorithms for the poisoning in the **training time**. In contrast, we consider the data poisoning problem, which happens before the victim trains a model.

## 3 Data Poisoning Attack to G-SSL

### 3.1 Problem setting

We consider the graph-based semi-supervised learning (G-SSL) problem. The input include labeled data $\boldsymbol{X}_l \in \mathbb{R}^{n_l \times d}$ and unlabeled data $\boldsymbol{X}_u \in \mathbb{R}^{n_u \times d}$, we define the whole features $\boldsymbol{X} = [\boldsymbol{X}_l; \boldsymbol{X}_u]$. Denoting the labels of $\boldsymbol{X}_l$ as $\boldsymbol{y}_l$, our goal is to predict the labels of test data $\boldsymbol{y}_u$. The learner applies algorithm $\mathcal{A}$ to predict $\boldsymbol{y}_u$ from available data $\{\boldsymbol{X}_l, \boldsymbol{y}_l, \boldsymbol{X}_u\}$. Here we restrict $\mathcal{A}$ to label propagation method, where we first generate a graph with adjacency matrix $\boldsymbol{S}$ from Gaussian kernel: $\boldsymbol{S}_{ij} = \exp(-\gamma \|\boldsymbol{x}_i - \boldsymbol{x}_j\|^2)$, where the subscripts $\boldsymbol{x}_{i(j)}$ represents the $i(j)$-th row of $\boldsymbol{X}$. Then the graph Laplacian is calculated by $\boldsymbol{L} = \boldsymbol{D} - \boldsymbol{S}$, where $\boldsymbol{D} = \text{diag}\{\sum_{k=1}^{n} \boldsymbol{S}_{ik}\}$ is the degree matrix. The unlabeled data is then predicted through energy minimization principle [2]

$$\min_{\hat{\boldsymbol{y}}} \frac{1}{2} \sum_{i,j} \boldsymbol{S}_{ij} (\hat{\boldsymbol{y}}_i - \hat{\boldsymbol{y}}_j)^2 = \hat{\boldsymbol{y}}^{\mathsf{T}} \boldsymbol{L} \hat{\boldsymbol{y}}, \quad \text{s.t.} \quad \hat{\boldsymbol{y}}_{:l} = \boldsymbol{y}_l. \tag{1}$$

The problem has a simple closed form solution $\hat{\boldsymbol{y}}_u = (\boldsymbol{D}_{uu} - \boldsymbol{S}_{uu})^{-1} \boldsymbol{S}_{ul} \boldsymbol{y}_l$, where we define $\boldsymbol{D}_{uu} = \boldsymbol{D}_{[0:u,0:u]}$, $\boldsymbol{S}_{uu} = \boldsymbol{S}_{[0:u,0:u]}$ and $\boldsymbol{S}_{ul} = \boldsymbol{S}_{[0:u,0:l]}$. Now we consider the attacker who wants to greatly change the prediction result $\boldsymbol{y}_u$ by perturbing the *training data* $\{\boldsymbol{X}_l, \boldsymbol{y}_l\}$ by small amounts $\{\Delta_x, \boldsymbol{\delta}_y\}$ respectively, where $\Delta_x \in \mathbb{R}^{n_l \times d}$ is the perturbation matrix , and $\boldsymbol{\delta}_y \in \mathbb{R}^{n_l}$ is a vector. This seems to be a simple problem at the first glance, however, we will show that the problem of finding optimal perturbation is often intractable, and therefore provable and effective algorithms are needed. To sum up, the problem have several degrees of freedom:

- **Learning algorithm**: Among all graph-based semi-supervised learning algorithms, we primarily focus on the label propagation method; however, we also discuss manifold regularization method in Appendix 4.2.
- **Task**: We should treat the regression task and classification task differently because the former is inherently a continuous optimization problem while the latter can be transformed into integer programming.
- **Knowledge of attacker**: Ideally, the attacker knows every aspect of the victim, including training data, testing data, and training algorithms. However, we will also discuss incomplete knowledge scenario; for example, the attacker may not know the exact value of hyper-parameters.
- **What to perturb**: We assume the attacker can perturb the label or the feature, but not both. We made this assumption to simplify our discussion and should not affect our findings.
- **Constraints**: We also assume the attacker has limited capability, so that (s)he can only make small perturbations. It could be measured $\ell_2$-norm or sparsity.

### 3.2 Toy Example

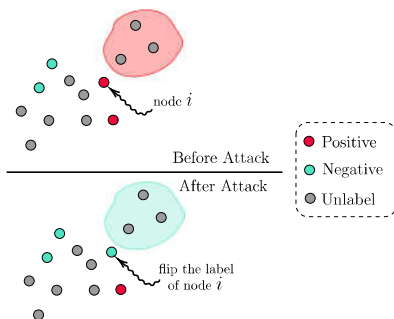

We show a toy example in Figure 1 to motivate the data poisoning attack to graph semi-supervised learning (let us focus on label propagation in this toy example). In this example, the shaded region is very close to node-$i$ and yet quite far from other labeled nodes. After running label propagation, all nodes inside the shaded area will be predicted to be the same label as node-$i$. That gives the attacker a chance to manipulate the decision of all unlabeled nodes in the shaded area at the cost of flipping just one node. For example, in Figure 1, if we change node-$i$'s label from positive to negative, the predictions in the shaded area containing three nodes will also change from positive to negative.

Besides changing the labels, another way to attack is to perturb the features $\boldsymbol{X}$ so that the graph structure $\boldsymbol{S}$ changes subtly (recall the graph structure is constructed based on pair-wise distances). For instance, we can change the features so that node $i$ is moved away from the shaded region, while more negative label points are moved towards the

Figure 1: We show a toy example that illustrates the main idea of the poisoning attack against SSL. By flipping just one training data from positive to negative, the prediction of the whole shaded area will be changed.

shaded area. Then with label propagation, the labels of the shaded region will be changed from positive to negative as well. We will examine both cases in the following sections.

### 3.3  A unified framework

The goal of poisoning attack is to modify the data points to maximize the error rate (for classification) or RMSE score (for regression); thus we write the objective as

$$\min_{\substack{\boldsymbol{\delta}_y \in \mathcal{R}_1 \\ \Delta_x \in \mathcal{R}_2}} -\frac{1}{2} \left\| g\Big( (\boldsymbol{D}'_{uu} - \boldsymbol{S}'_{uu})^{-1} \boldsymbol{S}'_{ul} (\boldsymbol{y}_l + \boldsymbol{\delta}_y) \Big) - h(\boldsymbol{y}_u) \right\|_2^2 \quad \text{s.t. } \{ \boldsymbol{D}', \boldsymbol{S}' \} = \mathrm{Ker}_\gamma (\boldsymbol{X}_l + \Delta_x). \quad (2)$$

To see the flexibility of Eq. (2) in modeling different tasks, different knowledge levels of attackers or different budgets, we decompose it into following parts that are changeable in real applications:

- $\mathcal{R}_1/\mathcal{R}_2$ are the constraints on $\boldsymbol{\delta}_y$ and $\Delta_x$. For example, $\mathcal{R}_1 = \{ \|\boldsymbol{\delta}_y\|_2 \leq d_{\max} \}$ restricts the perturbation $\boldsymbol{\delta}_y$ to be no larger than $d_{\max}$; while $\mathcal{R}_1 = \{ \|\boldsymbol{\delta}_y\|_0 \leq c_{\max} \}$ makes the solution to have at most $c_{\max}$ non-zeros. As to the choices of $\mathcal{R}_2$, besides $\ell_2$ regularization, we can also enforce group sparsity structure, where each row of $\Delta_x$ could be all zeros.
- $g(\cdot)$ is the task dependent squeeze function, for classification task we set $g(x) = \mathrm{sign}(x)$ since the labels are discrete and we evaluate the accuracy; for regression task it is identity function $g(x) = x$, and $\ell_2$-loss is used.
- $h(\cdot)$ controls the knowledge of unlabeled data. If the adversary knows the ground truth very well, then we simply put $h(\boldsymbol{y}_u) = \boldsymbol{y}_u$; otherwise one has to estimate it from Eq. (1), in other words, $h(\boldsymbol{y}_u) = \hat{\boldsymbol{y}}_u = g\big( (\boldsymbol{D}_{uu} - \boldsymbol{S}_{uu})^{-1} \boldsymbol{S}_{ul} \boldsymbol{y}_l \big)$.
- $\mathrm{Ker}_\gamma$ is the kernel function parameterized by $\gamma$, we choose Gaussian kernel throughout.
- Similar to $\boldsymbol{S}$, the new similarity matrix $\boldsymbol{S}'$ is generated by Gaussian kernel with parameter $\gamma$, except that it is now calculated upon poisoned data $\boldsymbol{X}_l + \Delta_x$.
- Although not included in this paper, we can also formulate targeted poisoning attack problem by changing min to max and let $h(\boldsymbol{y}_u)$ be the target.

There are two obstacles to solving Eq. 2, that make our algorithms non-trivial. First, the problem is naturally *non-convex*, making it hard to determine whether a specific solution is globally optimal; secondly, in classification tasks where our goal is to maximize the testing time error rate, the objective is *non-differentiable* under *discrete* domain. Besides, even with hundreds of labeled data, the domain space can be unbearably big for brute force search and yet the greedy search is too myopic to find a good solution (as we will see in experiments).

In the next parts, we show how to tackle these two problems separately. Specifically, in the first part, we propose an efficient solver designed for data poisoning attack to the regression problem under various constraints. Then we proceed to solve the discrete, non-differentiable poisoning attack to the classification problem.

### 3.4  Regression task, (un)known label

We first consider the regression task where only label poisoning is allowed. This simplifies Eq. (2) as

$$\min_{\|\boldsymbol{\delta}_y\|_2 \leq d_{\max}} \begin{cases} -\dfrac{1}{2} \left\| (\boldsymbol{D}_{uu} - \boldsymbol{S}_{uu})^{-1} \boldsymbol{S}_{ul} \boldsymbol{\delta}_y \right\|_2^2 & \text{(estimated label)} \quad (3a) \\[2ex] -\dfrac{1}{2} \left\| (\boldsymbol{D}_{uu} - \boldsymbol{S}_{uu})^{-1} \boldsymbol{S}_{ul} (\boldsymbol{y}_l + \boldsymbol{\delta}_y) - \boldsymbol{y}_u \right\|_2^2 & \text{(true label)} \quad (3b) \end{cases}$$

Here we used the fact that $\hat{\boldsymbol{y}}_u = \boldsymbol{K} \boldsymbol{y}_l$, where we define $\boldsymbol{K} = (\boldsymbol{D}_{uu} - \boldsymbol{S}_{uu})^{-1} \boldsymbol{S}_{ul}$. We can solve Eq. (3a) by SVD; it's easy to see that the optimal solution should be $\boldsymbol{\delta}_y = \pm d_{\max} \boldsymbol{v}_1$ and $\boldsymbol{v}_1$ is the top right sigular vector if we decompose $(\boldsymbol{D}_{uu} - \boldsymbol{S}_{uu})^{-1} \boldsymbol{S}_{ul} = \boldsymbol{U} \boldsymbol{\Sigma} \boldsymbol{V}^\intercal$. However, (3b) is less straightforward, in fact it is a non-convex trust region problem, which can be generally formulated as

$$\min_{\|\boldsymbol{z}\|_2 \leq d_{\max}} f(\boldsymbol{z}) = \frac{1}{2} \boldsymbol{z}^\intercal \boldsymbol{H} \boldsymbol{z} + \boldsymbol{g}^\intercal \boldsymbol{z}, \quad \boldsymbol{H} \text{ is indefinite.} \quad (4)$$

Our case (3b) can thus be described as $\boldsymbol{H} = -\boldsymbol{K}^\intercal \boldsymbol{K} \preceq 0$ and $\boldsymbol{g} = \boldsymbol{K}^\intercal (\boldsymbol{y}_u - \hat{\boldsymbol{y}}_u)$. Recently [26] proposed a sublinear time solver that is able to find a global minimum in $\mathcal{O}(M/\sqrt{\epsilon})$ time. Here

---

**Algorithm 1:** Trust region problem solver

---

**Data:** Vector $g$, symmetric indefinite matrix $H$ for problem $\min_{\|z\| \le 1} \frac{1}{2} z^\mathsf{T} H z + g^\mathsf{T} z$.
**Result:** Approximate solution $z^*$.

1  Initialize $z_0 = -0.5 \frac{g}{\|g\|}$ and step size $\eta$;
    /* Phase I: iterate inside sphere $\|z_t\| < 1$                          */
2  **while** $\|z_t\| < 1$ **do**
3     |   $z_{t+1} = z_t - \eta(H z_t + g)$;
4  **end**
    /* Phase II: iterate on the sphere $\|z_t\| = 1$                      */
5  $z_{t'} = z_t$;
6  **while** $t < \text{max\_iter}$ **do**
7     |   Choose $\alpha_{t'}$ by line search and do the following projected gradient descent on sphere;
8     |   $z_{t'+1} = \frac{z_{t'} - \alpha_{t'}(I_d - z_{t'} z_{t'}^\mathsf{T})(H z_{t'} + g)}{\|z_{t'} - \alpha_{t'}(I_d - z_{t'} z_{t'}^\mathsf{T})(H z_{t'} + g)\|}$;
9  **end**
10 **Return** $z_{\text{max\_iter}}$

---

we propose an asymptotic linear algorithm based purely on gradient information, which is stated in Algorithm 1 and Theorem 6. In Algorithm 1 there are two phases, in the following theorems, we show that the phase I ends within finite iterations, and phase II converges with an asymptotic linear rate. We postpone the proof to Appendix 1.

**Theorem 1** (Convergent). *Suppose the operator norm $\|H\|_{\text{op}} = \beta$, by choosing a step size $\eta < 1/\beta$ with initialization $z_0 = -\alpha \frac{g}{\|g\|}$, $0 < \alpha < \min(1, \frac{\|g\|^3}{|g^\mathsf{T} H g|})$. Then iterates $\{z_t\}$ generated from Algorithm 1 converge to the global minimum.*

**Lemma 1** (Finite phase I). *Since $H$ is indefinite, $\lambda_1 = \lambda_{\min}(H) < 0$, and $v_1$ is the corresponding eigenvector. Denote $a^{(1)} = a^\mathsf{T} v_1$ is the projection of any $a$ onto $v_1$, let $T_1$ be number of iterations in phase I of Algorithm 1, then:*

$$T_1 \le \log(1 - \eta\lambda_1)^{-1} \left[ \log\left(\frac{1}{\eta|g^{(1)}|} - \frac{1}{\eta\lambda_1}\right) - \log\left(\frac{-z_0^{(1)}}{\eta g^{(1)}} - \frac{1}{\eta\lambda_1}\right) \right]. \tag{5}$$

**Theorem 2** (Asymptotic linear rate). *Let $\{z_t\}$ be an infinite sequence of iterates generated by Algorithm 1, suppose it converges to $z^*$ (guaranteed by Theorem 1), let $\lambda_{H,\min}$ and $\lambda_{H,\max}$ be the smallest and largest eigenvalues of $H$. Assume that $z^*$ is a local minimizer then $\lambda_{H,\min} > 0$ and given $r$ in the interval $(r_*, 1)$ with $r_* = 1 - \min\left(2\sigma\bar\alpha\lambda_{H,\min}, 4\sigma(1-\sigma)\beta\frac{\lambda_{H,\min}}{\lambda_{H,\max}}\right)$, $\bar\alpha$, $\sigma$ are line search parameters. There exists an integer $K$ such that:*

$$f(z_{t+1}) - f(z^*) \le r\big(f(z_t) - f(z^*)\big)$$

*for all $t \ge K$.*

### 3.5 Classification task

As we have mentioned, data poisoning attack to classification problem is more challenging, as we can only *flip* an unnoticeable fraction of training labels. This is inherently a combinatorial optimization problem. For simplicity, we restrict the scope to binary classification so that $y_l \in \{-1, +1\}^{n_l}$, and the labels are perturbed as $\tilde{y}_l = y_l \odot \delta_y$, where $\odot$ denotes Hadamard product and $\delta_y = [\pm 1, \pm 1, \ldots, \pm 1]$. For restricting the amount of perturbation, we replace the norm constraint in Eq. (3a) with integer constraint $\sum_{i=1}^{n_l} \mathbb{I}_{\{\delta_y[i] = -1\}} \le c_{\max}$, where $c_{\max}$ is a user pre-defined constant. In summary, the final objective function has the following form

$$\min_{\delta_y \in \{+1, -1\}^{n_l}} -\frac{1}{2} \left\| g\big(K(y_l \odot \delta_y)\big) - (y_u \text{ or } \hat{y}_u) \right\|^2, \quad \text{s.t.} \quad \sum_{i=1}^{n_l} \mathbb{I}_{\{\delta_y[i] = -1\}} \le c_{\max}, \tag{6}$$

where we define $K = (D_{uu} - S_{uu})^{-1} S_{ul}$ and $g(x) = \text{sign}(x)$, so the objective function directly relates to error rate. Notice that the feasible set contains around $\sum_{k=0}^{c_{\max}} \binom{n_l}{k}$ solutions, making it almost impossible to do an exhaustive search. A simple alternative is greedy search: first initialize

$\boldsymbol{\delta}_y = [+1, +1, \ldots, +1]$, then at each time we select index $i \in [n_l]$ and try flip $\boldsymbol{\delta}_y[i] = +1 \rightarrow -1$, such that the objective function (6) decreases the most. Next, we set $\boldsymbol{\delta}_y[i] = -1$. We repeat this process multiple times until the constraint in (6) is met.

Doubtlessly, the greedy solver is myopic. The main reason is that the greedy method cannot explore other flipping actions that appear to be sub-optimal within the current context, despite that some sub-optimal actions might be better in the long run. Inspired by the bandit model, we can imagine this problem as a multi-arm bandit, with $n_l$ arms in total. And we apply a strategy similar to $\epsilon$-greedy: each time we assign a high probability to the best action but still leave non-zero probabilities to other "actions". The new strategy can be called *probabilistic method*, specifically, we model each action $\boldsymbol{\delta}_y = \pm 1$ as a Bernoulli distribution, the probability of "flipping" is $P[\boldsymbol{\delta}_y = -1] = \boldsymbol{\alpha}$. The new loss function is just an expectation over Bernoulli variables

$$\min_{\boldsymbol{\alpha}} \left\{ \mathcal{L}(\boldsymbol{\alpha}) := -\frac{1}{2} \mathop{\mathbb{E}}_{\boldsymbol{z} \sim \mathcal{B}(\boldsymbol{1}, \boldsymbol{\alpha})} \left[ \left\| g\big(\boldsymbol{K}(\boldsymbol{y}_l \odot \boldsymbol{z})\big) - (\boldsymbol{y}_u \text{ or } \hat{\boldsymbol{y}}_u) \right\|^2 \right] + \frac{\lambda}{2} \cdot \|\boldsymbol{\alpha}\|_2^2 \right\}. \tag{7}$$

Here we replace the integer constraint in Eq. 6 with a regularizer $\frac{\lambda}{2} \|\boldsymbol{\alpha}\|_2^2$, the original constraint is reached by selecting a proper $\lambda$. Once problem (7) is solved, we craft the actual perturbation $\boldsymbol{\delta}_y$ by setting $\boldsymbol{\delta}_y[i] = -1$ if $\boldsymbol{\alpha}[i]$ is among the top-$c_{\max}$ largest elements.

To solve Eq. (7), we need to find a good gradient estimator. Before that, we replace $g(x) = \text{sign}(x)$ with $\tanh(x)$ to get a continuously differentiable objective. We borrow the idea of "reparameterization trick" [27, 28] to approximate $\mathcal{B}(\boldsymbol{1}, \boldsymbol{\alpha})$ by a continuous random vector

$$\boldsymbol{z} \triangleq \boldsymbol{z}(\boldsymbol{\alpha}, \Delta_G) = \frac{2}{1 + \exp\left(\frac{1}{\tau}\left(\log \frac{\boldsymbol{\alpha}}{1 - \boldsymbol{\alpha}} + \Delta_G\right)\right)} - 1 \in (-1, 1), \tag{8}$$

where $\Delta_G \sim \boldsymbol{g}_1 - \boldsymbol{g}_2$ and $\boldsymbol{g}_{1,2} \overset{\text{iid}}{\sim} \text{Gumbel}(0, 1)$ are two Gumbel distributions. $\tau$ is the temperature controlling the steepness of sigmoid function: as $\tau \rightarrow 0$, the sigmoid function point-wise converges to a stair function. Plugging (8) into (7), the new loss function becomes

$$\mathcal{L}(\boldsymbol{\alpha}) := -\frac{1}{2} \mathop{\mathbb{E}}_{\Delta_G} \left[ \left\| g\big(\boldsymbol{K}(\boldsymbol{y}_l \odot \boldsymbol{z}(\boldsymbol{\alpha}, \Delta_G))\big) - (\boldsymbol{y}_u \text{ or } \hat{\boldsymbol{y}}_u) \right\|^2 \right] + \frac{\lambda}{2} \cdot \|\boldsymbol{\alpha}\|_2^2. \tag{9}$$

Therefore, we can easily obtain an unbiased, low variance gradient estimator via Monte Carlo sampling from $\Delta_G = \boldsymbol{g}_1 - \boldsymbol{g}_2$, specifically

$$\frac{\partial \mathcal{L}(\boldsymbol{\alpha})}{\partial \boldsymbol{\alpha}} \approx -\frac{1}{2} \frac{\partial}{\partial \boldsymbol{\alpha}} \left\| g\big(\boldsymbol{K}(\boldsymbol{y}_l \odot \boldsymbol{z}(\boldsymbol{\alpha}, \Delta_G))\big) - (\boldsymbol{y}_u \text{ or } \hat{\boldsymbol{y}}_u) \right\|^2 + \lambda \boldsymbol{\alpha}. \tag{10}$$

Based on that, we can apply many stochastic optimization methods, including SGD and Adam [29], to finalize the process. In the experimental section, we will compare the greedy search with our probabilistic approach on real data.

## 4 Experiments

In this section, we will show the effectiveness of our proposed data poisoning attack algorithms for regression and classification tasks on graph-based SSL.

### 4.1 Experimental settings and baselines

We conduct experiments on two regression and two binary classification datasets[1]. The meta-information can be found in Table 4. We use a Gaussian kernel with width $\gamma$ to construct the graph. For each data, we randomly choose $n_l$ samples as the labeled set, and the rest are unlabeled. We normalize the feature vectors by

Table 1: Dataset statistics. Here $n$ is the total number of samples, $d$ is the dimension of feature vector and $\gamma^*$ is the optimal $\gamma$ in validation. mnist17 is created by extracting images for digits '1' and '7' from standard mnist dataset.

| Name | Task | $n$ | $d$ | $\gamma^*$ |
|---|---|---|---|---|
| cadata | Regression | 8,000 | 8 | 1.0 |
| E2006 | Regression | 19,227 | 150,360 | 1.0 |
| mnist17 | Classification | 26,014 | 780 | 0.6 |
| rcv1 | Classification | 20,242 | 47,236 | 0.1 |

[1]Publicly available at https://www.csie.ntu.edu.tw/~cjlin/libsvmtools/datasets/

$x' \leftarrow (x - \mu)/\sigma$, where $\mu$ is the sample mean, and $\sigma$ is the sample variance. For regression data, we also scale the output by $y' \leftarrow (y - y_{\min})/(y_{\max} - y_{\min})$ so that $y' \in [0, 1]$. To evaluate the performance of label propagation models, for regression task we use RMSE metric defined as $\text{RMSE} = \sqrt{\frac{1}{n_u} \sum_{i=1}^{n_u} (y_i - \hat{y}_i)^2}$, while for classification tasks we use error rate metric. For comparison with other methods, since **this is the first work on data poisoning attack to G-SSL**, we proposed several baselines according to graph centrality measures. The first baseline is random perturbation, where we randomly add Gaussian noise (for regression) or Bernoulli noise (for regression) to labels. The other two baselines based on graph centrality scores are more challenging, they are widely used to find the "important" nodes in the graph. Intuitively, we need to perturb "important" nodes to attack the model, and we decide the importance by node degree or PageRank. We explain the baselines with more details in the appendix.

### 4.2 Effectiveness of data poisoning to G-SSL

In this experiment, we consider the white-box setting where the attacker knows not only the ground truth labels $\boldsymbol{y}_u$ but also the correct hyper-parameter $\gamma^*$. We thus apply our proposed label poisoning algorithms in Section 3.4 and 3.5 to attack regression and classification tasks, respectively. In particular, we apply $\ell_2$ constraint for perturbation $\boldsymbol{\delta}_y$ in the regression task and use the greedy method in the classification task. The results are shown in Figure 2, as we can see in this figure, for both

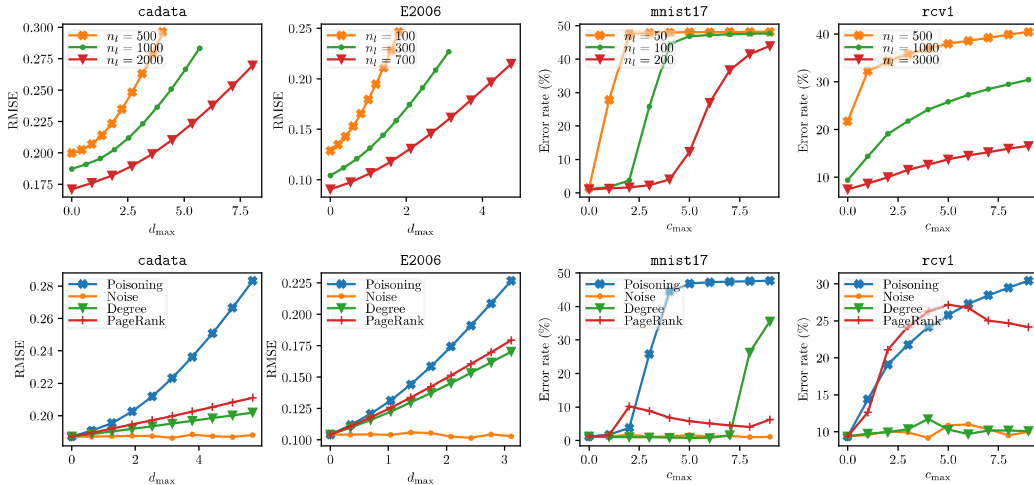

Figure 2: *Top row:* testing the effectiveness of poisoning algorithms on four datasets shown in Table (4). The left two datasets are regression tasks, and we report the RMSE measure. The right two datasets are classification tasks in which we report the error rate. For each dataset, we repeat the same attacking algorithm w.r.t. different $n_l$'s. *Bottom row:* compare our poisoning algorithm with three baselines (random noise, degree-based attack, PageRank based attack). We follow our convention that $d_{\max}$ is the maximal $\ell_2$-norm distortion, and $c_{\max}$ is the maximal $\ell_0$-norm perturbation.

regression and classification problems, small perturbations can lead to vast differences: for instance, on `cadata`, the RMSE increases from 0.2 to 0.3 when applied a carefully designed perturbation $\|\boldsymbol{\delta}_y\| = 3$ (this is very small compared with the norm of label $\|\boldsymbol{y}_l\| \approx 37.36$); More surprisingly, on `mnist17`, the accuracy can drop from 98.46% to 50% by flipping just 3 nodes. This phenomenon indicates that **current graph-based SSL, especially the label propagation method, can be very fragile to data poisoning attacks**. On the other hand, using different baselines (shown in Figure 2, bottom row), the accuracy does not decline much, this indicates that our proposed attack algorithms are more effective than centrality based algorithms.

Moreover, the robustness of label propagation is strongly related to the number of labeled data $n_l$: for all datasets shown in Figure 2, we notice that the models with larger $n_l$ tend to be more resistant to poisoning attacks. This phenomenon arises because, during the learning process, the label information propagates from labeled nodes to unlabeled ones. Therefore even if a few nodes are "contaminated" during poisoning attacks, it is still possible to recover the label information from other labeled nodes.

Hence this experiment can be regarded as another instance of "no free lunch" theory in adversarial learning [30].

## 4.3 Comparing poisoning with and without truth labels

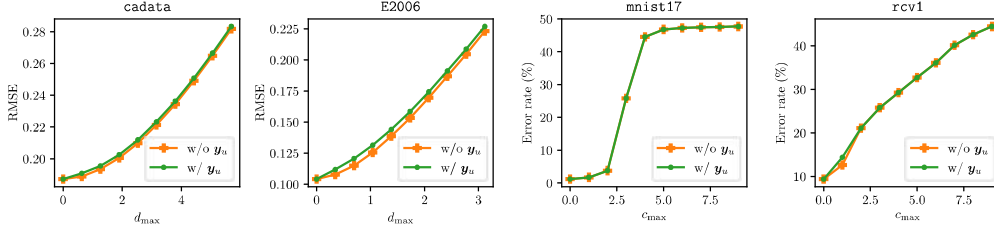

Figure 3: Comparing the effectiveness of label poisoning attack with and without knowing the ground truth labels of unlabeled nodes $\boldsymbol{y}_u$. Interestingly, even if the attacker is using the estimated labels $\hat{\boldsymbol{y}}_u$, the effectiveness of the poisoning attack does not degrade significantly.

We compare the effectiveness of poisoning attacks with and without ground truth labels $\boldsymbol{y}_u$. Recall that if an attacker does not hold $\boldsymbol{y}_u$, (s)he will need to replace it with the estimated values $\hat{\boldsymbol{y}}_u$. Thus we expect a degradation of effectiveness due to the replacement of $\boldsymbol{y}_u$, especially when $\hat{\boldsymbol{y}}_u$ is not a good estimation of $\boldsymbol{y}_u$. The result is shown in Figure 3. Surprisingly, we did not observe such phenomenon: for regression tasks on cadata and E2006, two curves are closely aligned despite that attacks without ground truth labels $\boldsymbol{y}_u$ are only slightly worse. For classification tasks on mnist17 and rcv1, we cannot observe any difference, the choices of which nodes to flip are exactly the same (except the $c_{\max} = 1$ case in rcv1). This experiment provides a valuable implication that hiding the ground truth labels cannot protect the SSL models, because the attackers can alternatively use the estimated ground truth $\hat{\boldsymbol{y}}_u$.

## 4.4 Comparing greedy and probabilistic method

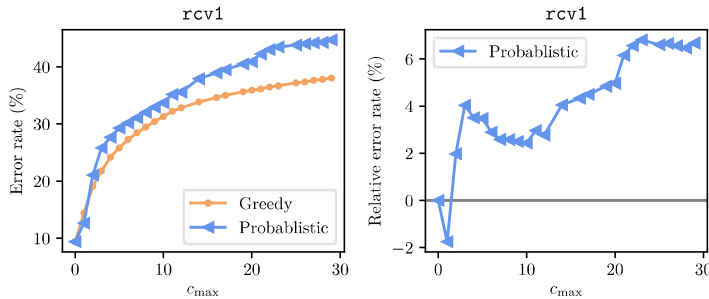

Figure 4: Comparing the relative performance of three approximate solvers to discrete optimization problem (6). For clarity, we also show the relative performance on the right (probabilistic − greedy).

In this experiment, we compare the performance of three approximate solvers for problem (6) in Section 3.5, namely greedy and probabilistic methods. We choose rcv1 data as oppose to mnist17 data, because rcv1 is much harder for poisoning algorithm: when $n_l = 1000$, we need $c_{\max} \approx 30$ to make error rate $\approx 50\%$, whilst mnist17 only takes $c_{\max} = 5$. For hyperparameters, we set $c_{\max} = \{0, 1, \ldots, 29\}$, $n_l = 1000$, $\gamma^* = 0.1$. The results are shown in Figure 4, we can see that for larger $c_{\max}$, greedy method can easily stuck into local optima and inferior than our probabilistic based algorithms.

## 4.5 Sensitivity analysis of hyper-parameter

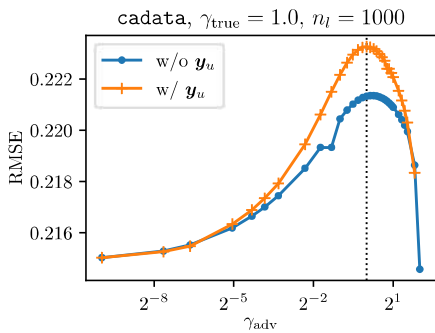

cadata, $\gamma_{\text{true}} = 1.0$, $n_l = 1000$

Figure 5: Experiment result on imperfect estimations of $\gamma^*$.

Since we use the Gaussian kernel to construct the graph, there is an important hyper-parameter $\gamma$ (kernel width) that controls the structure of the graph defined in (1), which is often chosen empirically by the victim through validation. Given the flexibility of $\gamma$, it is thus interesting to see how the effectiveness of the poisoning attack degrades with the attacker's imperfect estimation of $\gamma$. To this end, we suppose the victim runs the model at the optimal hyperparameter $\gamma = \gamma^*$, determined by validation, while the attacker has a very rough estimation $\gamma_{\text{adv}} \approx \gamma^*$. We conduct this experiment on cadata when the attacker knows or does not know the ground truth labels $\boldsymbol{y}_u$, the result is exhibited in Figure 5. It shows that when the adversary does not have exact information of $\gamma$, it will receive some penalties on the performance (in RMSE or error rate). However, it is entirely safe to choose a smaller $\gamma_{\text{adv}} < \gamma_{\text{truth}}$ because the performance decaying rate is pretty low. Take Figure 5 for example, even though $\gamma_{\text{adv}} = \frac{1}{8}\gamma_{\text{truth}}$, the RMSE only drops from 0.223 to 0.218. On the other hand, if $\gamma_{\text{adv}}$ is over large, the nodes become more isolated, and thus the perturbations are harder to propagate to neighbors.

## 5   Conclusion

We conduct the first comprehensive study of data poisoning to G-SSL algorithms, including label propagation and manifold regularization (in the appendix). The experimental results for regression and classification tasks exhibit the effectiveness of our proposed attack algorithms. In the future, it will be interesting to study poisoning attacks for deep semi-supervised learning models.

## Acknowledgement

Xuanqing Liu and Cho-Jui Hsieh acknowledge the support of NSF IIS-1719097, Intel faculty award, Google Cloud and Nvidia. Zhu acknowledges NSF 1545481, 1561512, 1623605, 1704117, 1836978 and the MADLab AF COE FA9550-18-1-0166.

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
