[Supplementary Material]

# A Proof of Convergence

We show that our gradient based nonconvex trust region solver is able to find a global minimum efficiently. First recall the objective function

$$f(z^*) = \min_{\|z\|_2 \leq 1} f(z) = \frac{1}{2}z^\mathsf{T} H z + g^\mathsf{T} z, \quad \lambda_{\min}(H) < 0. \tag{11}$$

Suppose $H$ has decomposition $\sum_{i=1}^{n} \lambda_i v_i v_i^\mathsf{T}$ and rank $\lambda_1 \leq \cdots \leq \lambda_i \leq 0 \leq \cdots \leq \lambda_n$. We only focus on "easy case": in which case $g^{(1)} = g^\mathsf{T} v_1 \neq 0$ and $v_1$ is the corresponding eigenvector of $\lambda_1 = \lambda_{\min}(H)$. In opposite, the hard case $g^{(1)} = 0$ is hardly seen in practice due to rounding error, and to be safe we can also add a small Gaussian noise to $g$. To see the structure of solution, suppose the solution is $z^*$, then by KKT condition, we can get the condition of global optima

$$(H + \lambda I_d)z^* + g = 0,$$
$$\lambda(1 - \|z^*\|_2) = 0, \tag{12}$$
$$H + \lambda I_d \succeq 0.$$

By condition $\lambda_1 < 0$, further if $g^{(1)} \neq 0$, then $\lambda \geq -\lambda_1$ and $\|z^*\| = 1$. Because $g^{(1)} \neq 0$, which implies $\lambda > -\lambda_1$. Immediately we know $z^* = -(H + \lambda I_d)^{-1}g$ and $\lambda$ is the solution of $\sum_{i=1}^{n}(\frac{g^{(i)}}{\lambda_i + \lambda})^2 = 1$.

As a immediate application of (12), we can conclude the following lemma:

**Lemma 2.** *When $g^{(1)} \neq 0$ and $\lambda_1 < 0$, among all stationary points if $s^{(1)}g^{(1)} \leq 0$ then $s = z^*$ is the global minimum.*

*Proof.* We proof by contradiction. Suppose $s$ is a stationary point and $s^{(1)}g^{(1)} \leq 0$, according to (12) if $s$ is not a global minimum then the third condition in (12) should be violated, implying that $\lambda_1 + \lambda < 0$. Furthermore, for stationary point $s$, we know the gradient of Lagrangian is zero: $(H + \lambda I_d)z^* + g = 0$. Projecting this equation onto $v_1$ we get

$$(\lambda_1 + \lambda)s^{(1)} + g^{(1)} = 0. \tag{13}$$

By condition $g^{(1)} \neq 0$ we know $s^{(1)} \neq 0$; multiply both sides of Eq. 13 by $s^{(1)}$ we get $s^{(1)}g^{(1)} > 0$, which is in contradiction to $s^{(1)}g^{(1)} \leq 0$. $\qquad\square$

We now consider the projected gradient descent update rule $z_{t+1} = \text{Prox}_{\|\cdot\|_2}(z_t - \eta\nabla f(z_t))$, with following assumptions:

**Assumption 1.** *(Bounded step size) Step size $\eta < 1/\beta$, where $\beta = \|H\|_{\text{op}}$.*

**Assumption 2.** *(Initialize) $z_0 = -\alpha\frac{g}{\|g\|}$, $0 < \alpha < \min\left(1, \frac{\|g\|^3}{|g^\mathsf{T} H g|}\right)$.*

Under these assumptions, we next show proximal gradient descent converges to global minimum

**Theorem 3.** *Under proximal gradient descent update: $z_{t+1} = \text{Prox}_{\|\cdot\|_2}(z_t - \eta\nabla f(z_t))$, and Assumption 1 if $z_t^{(i)}g^{(i)} \leq 0$ then $z_{t+1}^{(i)}g^{(i)} \leq 0$. Combining with Assumption 2 and Lemma 2, if $\lambda_1 < 0$, $g^{(1)} \neq 0$ then $z_t$ converges to a global minimum $z^*$.*

*Proof.* Notice the projection onto sphere will not change the sign of $z_{t+1}^{(i)}$, so:

$$\text{sign}\left(z_{t+1}^{(i)}g^{(i)}\right) = \text{sign}\left((1 - \eta\lambda_i)z_t^{(i)}g^{(i)} - \eta g^{(i)2}\right)$$

$\eta_t < 1/|\lambda_n|$ ensures $1 - \eta\lambda_i > 0$ for all $i \in [n]$. From Assumption 2 we know $z_0^{(i)}g^{(i)} = -\alpha\frac{g^{(i)2}}{\|g\|} \leq 0$, so $z_t^{(i)}g^{(i)} \leq 0$ for all $t$. We complete the proof by combining it with Lemma 2. $\qquad\square$

By careful analysis, we can actually divide the convergence process into two stages. In the first stage, the iterates $\{z_t\}$ stay inside the sphere $\|z_t\| < 1$; in the second stage the iterates stay on the unit ball $\|z_t\| = 1$. Furthermore, we can show that the first stage ends with finite number of iterations. Before that, we introduce the following lemma similar to [31] (but slightly different in that our problem has a norm constraint):

**Lemma 3.** *Considering the first stage, when iterates $\{z_t\}$ are inside unit sphere $\|z_t\|_2 \leq 1$, i.e. under the update rule $z_{t+1} = z_t - \eta \nabla f(z_t)$, and under assumption that $z_t^\mathsf{T} \nabla f(z_t) \leq 0$, we will have $z_t^\mathsf{T} H \nabla f(z_t) \geq \beta z_t^\mathsf{T} \nabla f(z_t)$ (recall we define $\beta$ as the operator norm of $H$).*

*Proof.* We first define $w_t^{(i)} = z_t^{(i)}/(-\eta g^{(i)})$, then by iteration rule $z_{t+1} = (I - \eta H)z_t - \eta g$, projecting both sides on $v_i$,

$$z_{t+1}^{(i)} = (1 - \eta \lambda_i)z_t^{(i)} - \eta g^{(i)},$$

dividing both sides by $-\eta g^{(i)}$, we get

$$w_{t+1}^{(i)} = (1 - \eta \lambda_i)w_t^{(i)} + 1, \tag{14}$$

solving this geometric series, we get:

$$w_t^{(i)} = (1 - \eta \lambda_i)^t \big(w_0^{(i)} - \frac{1}{\eta \lambda_i}\big) + \frac{1}{\eta \lambda_i}, \tag{15}$$

suppose at $t$-th iteration we have $w_t^{(i)} \geq w_{t+1}^{(i)}$, after plugging in Eq. (15) and noticing $0 < \eta \lambda_i < 1$

$$w_0^{(i)} - \frac{1}{\eta \lambda_i} \geq (1 - \eta \lambda_i)\big(w_0^{(i)} - \frac{1}{\eta \lambda_i}\big). \tag{16}$$

Furthermore, from Assumption 2 we know that $w_0^{(i)} = z_t^{(i)}/(-\eta g^{(i)}) = \alpha \frac{1}{\eta \|g\|_2} \geq 0$, if we have $w_0^{(i)} - \frac{1}{\eta \lambda_i} \leq 0$, equivalently $0 < \eta \lambda_i \leq 1/w_0^{(i)}$ then by Eq. (16) we know $1 - \eta \lambda_i \geq 1 \Leftrightarrow \eta \lambda_i \leq 0$ leading to a contradiction, so it must hold that

$$w_0^{(i)} - \frac{1}{\eta \lambda_i} > 0 \text{ and } 1 - \eta \lambda_i \leq 1.$$

At the same time, the eigenvalues are nondecreasing, $\lambda_j \geq \lambda_i$ for $j \geq i$, which means

$$1 - \eta \lambda_j \leq 1. \tag{17}$$

Also recalling the initialization condition implies $w_0^{(j)} = \alpha \frac{1}{\eta \|g\|_2} = w_0^{(i)}$, subtracting both sides by $1/\eta \lambda_j$ and noticing $\lambda_j \geq \lambda_i$

$$w_0^{(j)} - \frac{1}{\eta \lambda_j} = w_0^{(i)} - \frac{1}{\eta \lambda_j} \geq w_0^{(i)} - \frac{1}{\eta \lambda_i} > 0. \tag{18}$$

Combining Eq. (17) with Eq. (18), we can conclude that if $w_t^{(i)} \geq w_{t+1}^{(i)}$ holds, then such relation also holds for index $j > i$

$$w_0^{(j)} - \frac{1}{\eta \lambda_j} \geq (1 - \eta \lambda_j)\big(w_0^{(j)} - \frac{1}{\eta \lambda_j}\big) \Longleftrightarrow w_t^{(j)} \geq w_{t+1}^{(j)} \text{ for } j \geq i. \tag{19}$$

Consider at any iteration time $t$, suppose $i^* \in [n]$ is the smallest coordinate index such that $w_t^{(i^*)} \geq w_{t+1}^{(i^*)}$, and hence $w_t^i < w_{t+1}^i$ holds for all $i < i^*$. By Eq. (19) we know and $w_t^i \geq w_{t+1}^i$ for any $i \geq i^*$ (such a $i^*$ may not exist, but it doesn't matter). By analyzing the sign of $z_t$ we know:

$$\text{sign}\big(z_t^i\big(z_t^{(i)} - z_{t+1}^{(i)}\big)\big) = \text{sign}\big(w_t^{(i)}\big(w_t^{(i)} - w_{t+1}^{(i)}\big)\big) = \text{sign}\big(w_t^{(i)} - w_{t+1}^{(i)}\big), \forall i \in [n],$$

the second equality is true due to Eq. (14), we know $w_t^{(i)} > 0$ for all $i$ and $t$.

We complete the proof by following inequalities:

$$
\begin{aligned}
z_t^\mathsf{T} A \nabla f(z_t) &= \frac{1}{\eta} \sum_{i=1}^{i^*-1} \underbrace{\lambda_i z_t^{(i)}\big(z_t^{(i)} - z_{t+1}^{(i)}\big)}_{\leq 0} + \frac{1}{\eta} \sum_{i=i^*}^{n} \underbrace{\lambda_i z_t^{(i)}\big(z_t^{(i)} - z_{t+1}^{(i)}\big)}_{\geq 0} \\
&\geq \frac{\lambda_{i^*-1}}{\eta} \sum_{i=1}^{i^*-1} z_t^{(i)}\big(z_t^{(i)} - z_{t+1}^{(i)}\big) + \frac{\lambda_{i^*}}{\eta} \sum_{i=i^*}^{n} z_t^{(i)}\big(z_t^{(i)} - z_{t+1}^{(i)}\big) \\
&\geq \frac{\lambda_{i^*}}{\eta} \sum_{i=1}^{n} z_t^{(i)}\big(z_t^{(i)} - z_{t+1}^{(i)}\big) \\
&\geq \beta z_t^\mathsf{T} \nabla f(z_t).
\end{aligned}
\tag{20}
$$

Where the last inequality follows from assumption in this lemma. $\qquad\square$

By applying this lemma on the iterates $\{z_t\}$ that are still inside the sphere, we will eventually conclude that $\|z_t\|_2$ monotone increases. In fact, we have the following theorem:

**Theorem 4.** *Suppose $\{z_t\}$ is in the region $\|z_t\|_2 < 1$, such that proximal gradient update equals to plain GD: $z_{t+1} = z_t - \eta \nabla f(z_t)$, then under this update rule, $\|z_t\|$ is monotone increasing.*

*Proof.* We prove by induction. First of all, notice $\|z_{t+1}\|^2 = \|z_t\|^2 - 2\eta z_t^\mathsf{T} \nabla f(z_t) + \eta^2 \|\nabla f(z_t)\|^2$, to prove $\|z_{t+1}\|^2 \geq \|z_t\|^2$, it remains to show $z_t^\mathsf{T} \nabla f(z_t) \leq 0$. For $t = 0$ we note that

$$z_0^\mathsf{T} \nabla f(z_0) = \alpha^2 \frac{g^\mathsf{T} H g}{\|g\|^2} - \alpha \|g\| \leq 0, \tag{21}$$

where the last inequality follows from Assumption 2. Now suppose $z_{t-1}^\mathsf{T} \nabla f(z_{t-1}) \leq 0$ and by update rule $z_t = z_{t-1} - \eta \nabla f(z_{t-1})$ we know:

$$z_t^\mathsf{T} \nabla f(z_t) = z_{t-1}^\mathsf{T} \nabla f(z_{t-1}) - \eta \|\nabla f(z_{t-1})\|^2 - \underbrace{\eta z_{t-1}^\mathsf{T} A \nabla f(z_{t-1})}_{(1)}$$

$$+ \underbrace{\eta^2 \nabla f(z_{t-1})^\mathsf{T} A \nabla f(z_{t-1})}_{(2)}.$$

From Lemma 4 we know $(1) \geq \beta z_{t-1} \nabla f(z_{t-1})$ and recall $\beta$ is the operator norm of $A$, we have $(2) \leq \beta \|\nabla f(z_{t-1})\|^2$, combining them together:

$$z_t^\mathsf{T} \nabla f(z_t) \leq (1 - \beta \eta) z_{t-1}^\mathsf{T} \nabla f(z_{t-1}) - \eta(1 - \eta\beta) \|\nabla f(z_t)\|^2, \tag{22}$$

by choosing $\eta < 1/\beta$ we proved $z_t^\mathsf{T} \nabla f(z_t) \leq 0$.

Due to induction rule we know that $z_t^\mathsf{T} \nabla f(z_t) \leq 0$ holds for all $t$ and moreover, $\|z_{t+1}\|$ is monotone increasing. □

We can easily improve the results above, to show that phase I (where $\|z_t\| \leq 1$) will eventually terminate after finite number of iteration. This is formally described in the following proposition:

**Proposition 1.** *(Finite phase I) Assuming $\lambda_1 < 0$, suppose $t^*$ is the index that $\|z_{t^*}\| < 1$ and $\|z_{t^*+1}\| \geq 1$, then $t^*$ is bounded by:*

$$t^* \leq \log(1 - \eta\lambda_1)^{-1} \left[ \log\left( \frac{1}{\eta|g^{(1)}|} - \frac{1}{\eta\lambda_1} \right) - \log\left( \frac{-z_0^{(1)}}{\eta g^{(1)}} - \frac{1}{\eta\lambda_1} \right) \right]. \tag{23}$$

*Proof.* This directly follows from:

$$1 \leq \eta^2 g^{(1)2} w_{t^*+1}^{(1)2} = z_{t^*+1}^{(1)2} \leq \|z_{t^*+1}\|^2,$$

together with Eq. (15) immediately comes to Eq. (23). □

Lastly, it remains to show the converge rate in phase II, this is actually a standard manifold gradient descent problem

**Theorem 5.** *Let $\{z_t\}$ be an infinite sequence of iterates generated by line search gradient descent, then every accumulation point of $\{z_t\}$ is a stationary point of the cost function $f$.*

**Theorem 6.** *Let $\{z_t\}$ be an infinite sequence of iterates generated by line search gradient descent, suppose it converges to $z^*$. Let $\lambda_{H,\min}$ and $\lambda_{H,\max}$ be the smallest and largest eigenvalues of the Hessian at $z^*$. Assume that $z^*$ is a local minimizer then $\lambda_{H,\min} > 0$ and given $r$ in the interval $(r_*, 1)$ with $r_* = 1 - \min\left(2\sigma\bar{\alpha}\lambda_{H,\min}, 4\sigma(1 - \sigma)\beta \frac{\lambda_{H,\min}}{\lambda_{H,\max}}\right)$, there exists an integer $K$ such that:*

$$f(z_{t+1}) - f(z^*) \leq r\big(f(z_t) - f(z^*)\big),$$

*for all $t \geq K$.*

*Proof.* See Theorem 4.3.1 and Theorem 4.5.6 in [Absil et al., 2009]. □

To apply Theorem 6, we need to check $\lambda_{H,\min} > 0$. To do that we can directly calculate its value, by the definition of Riemanndian Hessian, we have

$$\text{Hess } f(\boldsymbol{x}) = \text{Hess}(f \circ \text{Exp}_{\boldsymbol{x}})(0_{\boldsymbol{x}}),$$
$$\langle \text{Hess } f(\boldsymbol{x})[\xi], \xi \rangle = \langle \text{Hess } (f \circ \text{Exp}_{\boldsymbol{x}})(0_{\boldsymbol{x}})[\xi], \xi \rangle. \tag{24}$$

Then for $\xi \in T_x \mathcal{M}$,

$$\langle \text{Hess } f(\boldsymbol{x})[\xi], \xi \rangle = \langle \text{Hess } (f \circ R_{\boldsymbol{x}})(0_{\boldsymbol{x}})[\xi], \xi \rangle = \frac{\mathrm{d}^2}{\mathrm{d}t^2} f(R_{\boldsymbol{x}}(t\xi)) \Big|_{t=0}, \tag{25}$$

we then expand $f(R_{\boldsymbol{x}}(t\xi))$ to,

$$f(R_{\boldsymbol{x}}(t\xi)) = \frac{\xi^{\mathsf{T}} \boldsymbol{H} \xi \cdot t^2 + \xi^{\mathsf{T}} \boldsymbol{H} \boldsymbol{x} \cdot t + \boldsymbol{x}^{\mathsf{T}} \boldsymbol{H} \boldsymbol{x}}{2\|\boldsymbol{x} + t\xi\|_2^2} + \frac{\boldsymbol{g}^{\mathsf{T}} \boldsymbol{x} + \boldsymbol{g}^{\mathsf{T}} \xi \cdot t}{\|\boldsymbol{x} + t\xi\|_2^2}. \tag{26}$$

By differentiating $t$ twice and set $t = 0$ (this can be done by software), we finally get

$$\langle \text{Hess } f(\boldsymbol{x})[\xi], \xi \rangle = -\boldsymbol{x}^{\mathsf{T}} \boldsymbol{H} \boldsymbol{x} + \xi^{\mathsf{T}} \boldsymbol{H} \xi - \boldsymbol{g}^{\mathsf{T}} \boldsymbol{x}. \tag{27}$$

Taking $\boldsymbol{x} = \boldsymbol{z}^*$ into above equation, we get

$$\langle \text{Hess } f(\boldsymbol{z}^*)[\xi], \xi \rangle = -\boldsymbol{z}^{*\mathsf{T}} \boldsymbol{H} \boldsymbol{z}^* + \xi^{\mathsf{T}} \boldsymbol{H} \xi - \boldsymbol{g}^{\mathsf{T}} \boldsymbol{z}^*$$

On the other hand, by optimal condition (12), we have:

$$\boldsymbol{z}^{*\mathsf{T}}(\boldsymbol{H} + \lambda \boldsymbol{I}_d)\boldsymbol{z}^* + \boldsymbol{z}^{*\mathsf{T}} \boldsymbol{g} = 0 \implies -\boldsymbol{z}^{*\mathsf{T}} H \boldsymbol{z}^* - \boldsymbol{g}^{\mathsf{T}} \boldsymbol{z}^* = \lambda, \tag{28}$$

so $\langle \text{Hess} f(\boldsymbol{z}^*)[\xi], \xi \rangle = \lambda + \xi^{\mathsf{T}} \boldsymbol{H} \xi \geq \lambda + \lambda_1 \overset{!}{>} 0$. Where $\overset{!}{>}$ is guaranteed by gradient condition in (12):

$$(\lambda_1 + \lambda)\boldsymbol{z}^{*(1)} + \boldsymbol{g}^{(1)} = 0, \tag{29}$$

in "easy-case", $\boldsymbol{g}^{(1)} \neq 0$, so $\lambda_1 + \lambda \neq 0$ and Hessian condition in (12) can be improved to $\lambda_1 + \lambda > 0$.

Based on above discussion, we know $\lambda_{H,\min} \geq \lambda + \lambda_1 > 0$ and $\lambda_{H,\max} \leq \lambda + \lambda_n$.

## B   Supplementary Experiments on Trust Region Solver

We sample an indefinite random matrix by $\boldsymbol{H} = \boldsymbol{B}\boldsymbol{B}^{\mathsf{T}} - \lambda \boldsymbol{I}_n$, where $\boldsymbol{B} \in \mathbb{R}^{n \times (n-1)}$ and $\boldsymbol{B}_{ij} \overset{\text{iid}}{\sim} \mathcal{N}(0, 1)$, obviously $\lambda_{\min}(\boldsymbol{H}) = \lambda_1 = -\lambda$. Afterwards we sample a vector $\boldsymbol{g}$ by $\boldsymbol{g}_i \overset{\text{iid}}{\sim} \mathcal{N}(0, 1)$. it is totally fine to ignore the hard case, because the probability is zero. By changing the value of $\lambda$ in $\{10, 30, 50, 70, 90, 110\}$, we plot the function value decrement with respect to number of iterations in Figure 6(left). As we can see, the iterates first stay inside of the sphere (phase I) for a few iterations and then stay on the boundary (phase II). To inspect how $\lambda$ changes the duration of phase I, we then plot the number of iterations it takes to reach phase II, under different $\lambda$ values shown in Figure 6(right). Recall in (23), number of iterations is bounded as a function of $\lambda$, which can be further simplified to:

$$t^* \leq \frac{\log(1 + \frac{\lambda}{|\boldsymbol{g}^{(1)}|})}{\log(1 + \eta\lambda)} = \frac{\log(1 + c_1\lambda)}{\log(1 + c_2\lambda)}, \tag{30}$$

where we set $\boldsymbol{z}_0^{(1)} = 0$ to simplify the formula. By fitting the data point with function $T(\lambda) = \frac{\log(1+c_1\lambda)}{\log(1+c_2\lambda)}$, we find our bounds given by Lemma 6 is quite accurate.

## C   Baselines

There are three baselines included in the experiments, namely random noise, degree-based poisoning and PageRank-based poisoning. The first baseline, random noise, is the simplest one. For continuous label, the perturbation is created by $\boldsymbol{\delta}_{\boldsymbol{y}} = d_{\max} \frac{\epsilon}{\|\epsilon\|_2}$ with $\epsilon \sim \mathcal{N}(0, 1)$; for discrete label, we randomly choose $c_{\max}$ indices $\{k_1, k_2, \ldots, k_{c_{\max}}\}$ from $\{1, 2, \ldots, n\}$ and then set $\boldsymbol{\delta}_{\boldsymbol{y}}[k_i] = -1$.

As to degree-based poisoning, we first calculate the degree vector $\deg[i] = \sum_{j=1}^n \boldsymbol{S}_{ij}$ of all nodes, then for continuous label we load the perturbation weighted by degree, i.e. $|\boldsymbol{\delta}_{\boldsymbol{y}}[i]| =$

Figure 6: *Left*: Trust region experiment, we use solid lines to indicate iterations inside the sphere and dash lines to indicate iterations on the sphere. By changing $\lambda$ we can modify the function curvature. *Right*: #Iteration it takes to reach sphere under different $\lambda$'s, we also fit the curve by model $T = \frac{\log(1+c_1\lambda)}{\log(1+c_2\lambda)}$ derived in Eq. (23).

$d_{\max}\sqrt{\frac{\deg^2[i]}{\sum_{j=1}^n \deg^2[j]}}$, and the sign of $\boldsymbol{\delta_y}[i]$ is determined by the gradient of loss on $\boldsymbol{\delta_y}$ at $\boldsymbol{\delta_y} = \mathbf{0}$. Specifically:

$$\text{sign}(\boldsymbol{\delta_y}) = \text{sign}\left(\frac{\partial L(\boldsymbol{\delta_y})}{\partial \boldsymbol{\delta_y}}|_{\boldsymbol{\delta_y}=0}\right),$$

this makes sure that the direction is good enough to increase the prediction loss of SSL models. For discrete label, we simply choose the largest $c_{\max}$ training labels to flip: $\boldsymbol{\delta_y}[i] = -1$ if and only if node-$i$ has many neighboring nodes. This is to maximize the influence of perturbations.

Similarly, we can also a PageRank based poisoning attack, the only difference is that we use PageRank to replace the degree score.

## D  Supplementary Experiments on Data Poisoning Attacks

In this section, we design more experiments on other cases that are not able show up in the main text. The problem settings and datasets are the same as previous experiments.

### D.1  Sparse and group sparse constraint

In reality, the adversary may only be able to perturb very small amount of data points, this requirement renders sparse constraint. In specific, we consider

$$\mathcal{R}_1 = \{\|\boldsymbol{\delta_y}\|_0 \leq c_{\max} \text{ and } \|\boldsymbol{\delta_y}\|_2 \leq d_{\max}\},$$
$$\mathcal{R}_2 = \left\{\sum_{i=1}^n \mathbb{I}\{\Delta_x[i,:] \neq \mathbf{0}\} \leq c_{\max}\right\}. \tag{31}$$

The constraint on $\mathcal{R}_2$ implies that only a limited number of data can be perturbed, so we enforce a row-wise group sparsity. Both $\mathcal{R}_1$ and $\mathcal{R}_2$ can be added to regression/classification tasks, below we take regression task as an example to show the effectiveness.

For $\mathcal{R}_1$, the optimization problem is essentially a sparse PCA problem

$$\min_{\boldsymbol{\delta_y}} -\frac{1}{2}\left\|(\boldsymbol{D}_{uu} - \boldsymbol{S}_{uu})^{-1}\boldsymbol{S}_{ul}\boldsymbol{\delta_y}\right\|_2^2$$
$$\texttt{s.t.} \ \|\boldsymbol{\delta_y}\|_0 \leq c_{\max}, \ \|\boldsymbol{\delta_y}\|_2 \leq d_{\max}. \tag{32}$$

For this kind of problem, many efficient solvers were proposed during the past decades, including threshold method [32], LASSO based method [33], or by convex relaxation [34].

In order to solve the sparse PCA problem, we adopt the LASSO based sparse PCA solver [33]. We conduct the experiment on `cadata` and E2006 data, then plot the sparsity (measured by #nnz of

Figure 7: The effectiveness of $\ell_0/\ell_2$-mixed constraints in finding the sparse perturbations $\boldsymbol{\delta}_y$. For `cadata`, we set $n_l = 1000$, while for `E2006` data $n_l = 300$. The RMSE results of dense solutions (3a) are marked with red dashed lines.

Figure 8: We ranked the perturbations $\boldsymbol{\Delta}_l$ by their $\ell_2$-norm, to see the decay rate of distortion.

$\boldsymbol{\delta}_y$) and corresponding RMSE in Figure 7. For comparison, we also include the RMSE when no $\ell_0$ sparsity constraint is enforced. Interestingly, we observe that the RMSE increases rapidly as $\boldsymbol{\delta}_y$ is relatively sparse, and later it gradually stabilizes before reaching the same RMSE of dense solution. That is to say, when attackers have constraint on the maximal number of perturbation they could make, our sparse PCA based solution is able to make a good trade-off between sparsity and RMSE.

For $\mathcal{R}_2$, the har dconstraint is replaced with a group LASSO regularizer $\lambda \sum_{i=1}^{n_l} \|\boldsymbol{\Delta}_l[i,:]\|_2$ and we use proximal gradient descent to solve the optimization problem. By changing the hyper-parameter $\lambda$ we can indirectly change the group sparsity of $\Delta_x$. As above, we run the experiment on `mnist17` data, result is shown in Figure 8. We visualize the top-3 most distorted images in Figure 9.

### D.2 Data poisoning attack on manifold regularization model

Apart from label propagation model for semi-supervised learning, our method can also be seamlessly applied to manifold regularization method [4]. Manifold regularization based SSL solves the following optimization problem

$$f^* = \operatorname*{arg\,min}_{f \in \mathcal{F}} \sum_{i=1}^{n_l} \ell\big(f(\boldsymbol{x}_i), \boldsymbol{y}_i\big) + \lambda \|f\|^2 + \beta \sum_{i,j=1}^{n_l+n_u} \boldsymbol{S}_{ij}\big(f(\boldsymbol{x}_i) - f(\boldsymbol{x}_j)\big)^2, \qquad (33)$$

where $\mathcal{F}$ is the set of model functions. $(\boldsymbol{x}_i, \boldsymbol{y}_i)$ is the $i$-th data pair in $(\boldsymbol{X}, \boldsymbol{y})$, $\ell(\hat{y}, y)$ is the loss function. The model family $\mathcal{F}$ ranges from linear models $f(\boldsymbol{x}) = \boldsymbol{w}^\intercal \boldsymbol{x}$ to very complex deep neural networks. If we limit our scope to the linear case, then (33) has a closed form solution:

$$\boldsymbol{w}^* = (\boldsymbol{X}_l^\intercal \boldsymbol{X}_l + \lambda \boldsymbol{I} + \beta \boldsymbol{X}^\intercal \boldsymbol{L} \boldsymbol{X})^{-1} \boldsymbol{X}_l^\intercal \boldsymbol{y}_l = \boldsymbol{P} \boldsymbol{y}_l \qquad (34)$$

(a)                          (b)                          (c)

$\|\delta_x\| = 3.84$          $\|\delta_x\| = 2.64$          $\|\delta_x\| = 2.47$

Figure 9: Visualize the original image $\boldsymbol{X}_l$ (top left), distortion $\boldsymbol{\Delta}_l$ (top right) and perturbed image $\boldsymbol{X}_l + \boldsymbol{\Delta}_l$ (bottom). In fact, only the most distorted image (a) is visually different from its original appearance.

where $\boldsymbol{X}_l$ is the feature matrix of all labeled nodes, $\boldsymbol{X}$ is the feature matrix of labeled and unlabeled nodes, $\boldsymbol{L}$ is the graph Laplacian.

Manifold regularization term in (34) enforces two nodes that are close to each other (i.e. large $\boldsymbol{S}_{ij}$) to hold similar labels, and that is similar to the objective of label propagation. This motivates us to extend our algorithms for attacking label propagation to attack manifold regularization based SSL. As an example, we discuss poisoning attack to manifold regularization model for regression task, where the problem can be formulated as

$$\min_{\|\boldsymbol{\delta}_y\| \leq d_{\max}} -\frac{1}{2}\|\boldsymbol{X}_u \boldsymbol{P}(\boldsymbol{y}_l + \boldsymbol{\delta}_y) - \boldsymbol{y}_u\|_2^2. \tag{35}$$

Clearly, Eq. (35) is again a non-convex trust region problem, and we can apply our trust region problem solver to it. For experiment, we take regression task on `cadata` as an example, different from label propagation, manifold regularization learns a parametric model $f_{\boldsymbol{w}}(\boldsymbol{x})$ that is able to generalize to unseen graph. So for manifold regularization we can do label poisoning attack in both transductive and inductive settings. The experiment result is shown in Figure 10.

Figure 10: Experiment result of manifold regularization on `cadata`, here we set $n_l = 500$, $n_u = 3500$ and the rest $n_g = 4000$ data are used for inductive learning.

In this experiment, we do both transductive setting, using the test set $\{\boldsymbol{X}_u, \boldsymbol{y}_u\}$ as in label propagation, and inductive setting, on a brand new set $\{\boldsymbol{X}_{\text{ind}}, \boldsymbol{y}_{\text{ind}}\}$ that never been accessed in training stage. We can see that for both settings the label poisoning attack algorithm has equally good performance.