[Reviews · NeurIPS 2019]

Reviewer 1



As the topic of the this work is very interesting, I'm not frustrated by the presentation, which has a large space for improvement. First, the motivation of the work is not so clear. Why is that an important concern to be able to poison a G-SSL model? The three references [1-3] provided seem at least a decade ago. Are they widely applied in modern security/privacy sensitive applications? Are there recent works bringing these techniques into consideration? Are the proposed methods applicable to recent techniques as well? Answering these questions may help to justify the motivation better. The explanation of G-SSL and the formulation in 3.1 is very confusing. In particular, we need to understand what the subscriptions i,j, iterate over for. For example, in (1), do they loop over all labeled data? unlabeled data? or one labeled one unlabeled? Also, the description "the unlabeled data is then predicted through energy minimization principle" doesn't seem to bound any variables in (1) to the "unlabeled data". For the formulation, capital $X_l$ and $X_u$ seem to indicate two sets of data points, rather than two data points. Then I cannot make sense of $X_l+\Delta_x$ in (2) (p3, L134). This confusion makes me unable to make sense about the threat model for the attack. Does it mean all data can be manipulated as long as the sum of the perturbation is small enough? or only a small group of instances can be perturbed? If the later, how these groups are chosen? The experiments also have many presentation issues. The datasets are not explained at all, except mnist17. Although a reference is provided in the footnote, we require the paper to be self-contained. In Fig 2, different $n_l$ are used for comparison, but not explained. I first find this notion in 3.5, in the summation of the constraint, without explanation. Does it mean the total number of labeled data? Despite these presentation issues, I'm concerned about how should I interpret the hyper-parameters such as d_max and RMSE. Sure, RMSE 0.2 increases to 0.3. But doesn't mean very bad? Some real examples, and qualitative analysis may clarify this concern. Last but not least, it's questionable how generic is the proposed unified framework. It seems to be applicable to only a specific form of learning algorithms defined in 3.1. These algorithms seem to require to construct a adjacency matrix from kernels, which may not scale well to large datasets? In Sec 4, it seems that datasets of more than 20,000 nodes are used, but I don't see details about how the G-SSL methods can handle them. Are they unlabeled data or labeled data? If my understanding of n_l is correct, then it seems that the G-SSL approaches can handle up to 2000 labeled nodes, which then can make sense. Again, all these details are very important to evaluate this work, but missing. I do not check the proofs of the theorems line-by-line, so I have no comments over them. Overall, I think this paper needs more work to resolve the clarity issue, and justify the motivation better. Some minor issues: Thm 2 is refering a theorem ??. Shouldn't the input for MNIST be of dimension 28*28=784 (rather than 780 in Table 4.1)? Sec 4.4 says three approximate solvers are compared, but it's actually two presented. PageRank should be cited.

Reviewer 2



This paper proposes a unifying objective function for the effectiveness of a data poisoning attack to graph-based semi-supervised learning (G-SSL). Different than the other works in the literature, the attack is considered to take place during training, so the training data is poisoned. Although the problem is discussed in a fair level of generality in terms of the components of the trained data to be changed, the authors restrict the attention to cases where the ‘outcome’ component of the training data is changed. The authors then propose two different algorithms to find the best attack under given constraints according to the objective function — one algorithm for regression where real-valued outcomes are assumed and the other algorithm for classification where the outcomes are discrete (in particular, binary for this work). While the first algorithm guarantees convergence and has a linear converge rate, the second algorithm is shown to perform well empirically. The paper is well written. As a non-expert, the motivation, the problem definition, the proposed methods, and the discussion along the presentation of the results were clear to me as a non-expert reader. Taking granted the authors’ claim that the a new problem is studied, the extent of experiments is satisfying. The unifying approach to formulate the problem of finding the best attach is also an important contribution.

Reviewer 3



In this paper, the authors propose a framework for data poisoning attack to graph-based semi-supervised problems. The novelty of this paper is that the data poisoning attack is introduced during training time. For label poisoning to regression task, the authors propose a gradient based algorithm to solve this nonconvex trust region problem, which can converge to a global minimum in linear rate. For label poisoning attack to classification task, which is an NP-hard integer programming problem, the authors adopt stochastic gradient decent optimization algorithm by leveraging Monte Carlo sampling to get gradient estimator. The authors conducted the experiments quite thoroughly, which show the effectiveness of the proposed methods. In general, the proposed algorithm is well motivated and mathematically convincing, and the paper is sufficiently well written. The authors give detailed theoretical analysis, and the proof is neat.

Reviewer 4



I thank the authors for the response. My score remains unchanged after reading all reviews and the response. Significance: The problem of data poisoning is important and inherently tied with algorithmic robustness. The paper initiates a new line of research by studying data poisoning in the context of semi-supervised learning. The work is significant from this perspective, and some follow-up works on this topic are expected. However, the authors fail to provide real scenarios, where a study on data poisoning in a semi-supervised setup would be useful. Originality: Although the problem is somewhat new, the novelty of the methods / theory is not clear. The unified optimisation problem in eqn (2) is certainly interesting, and so is its extension to regularised semi-supervised formulation in the appendix. Two algorithms are proposed: - for regression, it is shown that the problem is non-convex and a trust region solver is presented along with convergence guarantees. There exist several other methods to solve similar non-convex problems. It is not clear how the analysis is different / more challenging when compared to existing theory. - for classification, there is no theory and the proposed approach is simply greedy / epsilon-greedy. The challenges, if any, for extending such methods to the poisoning scenario is not clear Quality and Clarity: Apart from concerns related to originality, the quality of the work (both theory and numerics) is good. The paper is clearly written in most parts, but contains several typos throughout (see below) Minor comments: - line167: IF H = - K'K, shouldn't it be negative definite (all eigenvalues non-positive) instead of indefinite? - line168: M, epsilon undefined variables - Thm 2 statement: reference to Thm 1 missing - line236: "for classification" - Fig 4: There are 2 algorithms, not 3

[Author Response · NeurIPS 2019]

We would like to thank all reviewers for your helpful suggestions.

Response to Reviewer #1

We are sorry about the confusing descriptions in this submission, in the updated version we will check every symbol to make sure they are clear. In what follows we answer your questions to help clarify the notations and experiments.

1. "motivation not clear": G-SSL and specifically label propagation used to be an important semi-supervised learning algorithm and it is still not outdated today: for image/text data we can apply the deep feature to construct a graph and propagate the labels to unlabeled data. Neo4j, a industrial level machine learning library even contains a build-in module for label propagation and it has already been applied in social network [Speriosu], Pharmacy [Zhang], etc. Furthermore, G-SSL algorithms are still widely used in social network or recommender systems, where malicious users can easily create fake account or inject fake information into nodes. Lastly, the clean math of G-SSL provides provides a good starting point to generalize adversarial machine learning to other SSL methods, including deep SSL. We will try to include more real-world examples in the revised version.

2. "$X_l$ and $X_u$": following the standard notations, we use $X \in \mathbb{R}^{n \times d}$ to represent the feature matrix for the whole data, where each row of $X$ is a data point. The feature matrix is composed of labeled part and unlabeled part. We simply set $X = [X_l; X_u]$, i.e. the first $n_l$ rows of $X$ are labeled data and the following $n_u$ rows are unlabeled data, and $n = n_l + n_u$. With this in mind, it might be easier to understand $X_l + \Delta_x$—this is the feature of labeled data after perturbation by matrix $\Delta_x \in \mathbb{R}^{n_l \times d}$, which means the feature perturbation can be done to all labeled instances. Moreover, by applying group sparsity constraints on $\Delta_x$ (consider each row as a group), our algorithm can conduct perturbation only to a small fraction of instances to greatly change the results.

3. "subscriptions $i$, $j$": each row of $X$ is a data point. $x_i$ and $x_j$ are the $i$-th and $j$-th row of the matrix $X$ so $i$ and $j$ are actually the index of data ($i, j \in \{1, 2, \ldots, n\}$), and they are iterated within the whole dataset (including both labeled and unlabeled data). However, the constraint in Eq(1) indicates that $\hat{y}$ are fixed for the labeled part, so the free variables in (1) are the unlabeled part of $\hat{y}$.

4. "Datasets not explained": we will add more descriptions of datasets in the next version. Specifically, mnist and rcv1 are widely used datasets to test non-deep learning models such as SVM/logitstic regression/tree based models, cadata/E2006 are regression datasets for predicting the house price/stock volatility. All of them are publically available at libsvm repository.

5. "Scalability of our algorithm": please note that the computation overhead mainly lies in generating the kernel matrix $S$ in Eq(3), which is $\mathcal{O}(n^2 d)$, computing this matrix for 20,000 nodes is certainly doable. For large data, one can construct a $k$-NN graph efficiently by efficient similarity search tools, such as Faiss (by Facebook). $S$ will be sparse so the computations will be efficient even for very large datasets.

6. "Dimension of MNIST17 should be 784 rather than 780": 4 in 784 pixels are zero in every image, so we remove these four dimensions in the experiment. Removing them will not affect the final results.

(Speriosu) Twitter polarity classification with label propagation over lexical links and the follower graph, in *EMNLP* 2011.

(Zhang) Label Propagation Prediction of Drug-Drug Interactions Based on Clinical Side Effects, in *Scientific Reports* 2015.

Response to Reviewer #2

Thanks for your feedback! We will correct typos and revise our submission according to your suggestions.

Response to Reviewer #3

Regarding the scope of the paper: The main text of our paper is mainly on poisoning label propagation method. We further show how to poison another widely used SSL method – manifold regularization in Appendix 4.2.

Regarding the datasets: the datasets used in the experiment are all real world datasets (downloaded from libsvm data repository) and are widely used for benchmarking classification and regression models. We will provide more details about these datasets in the revised version of the paper.

Response to Reviewer #4

Regarding the novelty of theory: Our main theoretical contribution lies in the efficient trust region solver in the nonconvex case. To our knowledge, prior work is limited to trust region with cubic regularization [Carmon and Duchi], which is different from our setting (a $\ell_2$-norm constraint), and their work cannot be directly applied to our data poisoning problem. Furthermore, earlier trust region solvers (e.g., Dogleg or Steihaug's method) can only get an approximate solution or local minimizer, while our algorithm is guaranteed to return the global minimizer.

Regarding the novelty of algorithm: Finding good "label fliping" operations via probabilistic method is novel. We agree that this method has no theoretical guarantee, however we show empirically that this epsilon-greedy solver successfully avoids some sub-optimal solutions, and beats the greedy method by a significant margin.

Regarding a unified framework: For completeness, we also include manifold regularization method in the appendix. It is certainly preferable to design a framework that include all semi-supervised learning approaches. Regarding to this limitation, we do not intend to solve it completely in this submission and we appeal for follow-up works to give a better answer.

(Carmon and Duchi) Gradient Descent Efficiently Finds the Cubic-Regularized Non-Convex Newton Step, in *ArXiv* 2016.

[Meta-Review · NeurIPS 2019]

I recommend acceptance of this article subject to some corrections. The novelty of considering data poisoning in GSSL can be interesting for the NeurIPS community. The results presented here are founded by theoretical results and experiments are conducted. It opens new lines of research and can trigger new results. Because the paper has several weak points, we insist on the importance to handle the following comments. 1/ The quality of writing is not at the expected level and some improvements have to be done before publishing. However, the overall structure of the presentation is acceptable. Correction of typos and a better explanation of the notations can be done. This issue could be solved without drastically changing the paper. 2/ The motivations behind data poisoning in GSSL was not clearly exposed. Moreover, the paper has not proposed counter-measures as it would be expected. Some elements about the former were given in the rebuttal. 3/ The related work has no deep exposition of previous results on robustness and stability of GSSL methods that could be useful to understand and compare with the work done in this paper.